# Exploring microbial diversity and biosynthetic potential in zoo and wildlife animal microbiomes

Georges P. Schmartz[1,7], Jacqueline Rehner [2,7], Miriam J. Schuff[2], Leidy-Alejandra G. Molano [1], Sören L. Becker [2], Marcin Krawczyk [3], Azat Tagirdzhanov [1,4], Alexey Gurevich[4,5], Richard Francke[6], Rolf Müller [4], Verena Keller[1,7] & Andreas Keller [1,4,7] ✉

Understanding human, animal, and environmental microbiota is essential for advancing global health and combating antimicrobial resistance (AMR). We investigate the oral and gut microbiota of 48 animal species in captivity, comparing them to those of wildlife animals. Specifically, we characterize the microbiota composition, metabolic pathways, AMR genes, and biosynthetic gene clusters (BGCs) encoding the production of specialized metabolites. Our results reveal a high diversity of microbiota, with 585 novel species-level genome bins (SGBs) and 484 complete BGCs identified. Functional gene analysis of microbiomes shows diet-dependent variations. Furthermore, by comparing our findings to wildlife-derived microbiomes, we observe the impact of captivity on the animal microbiome, including examples of converging microbiome compositions. Importantly, our study identifies AMR genes against commonly used veterinary antibiotics, as well as resistance to vancomycin, a critical antibiotic in human medicine. These findings underscore the importance of the 'One Health' approach and the potential for zoonotic transmission of pathogenic bacteria and AMR. Overall, our study contributes to a better understanding of the complexity of the animal microbiome and highlights its BGC diversity relevant to the discovery of novel antimicrobial compounds.

Microorganisms and the microbiomes they shape wield considerable influence on broader ecological dynamics despite their small scale. While much attention in clinical microbial research focuses on human pathogens and associated microbiomes, there is a growing recognition of the interconnectedness among animal, human, and environmental health, underscored by the 'One Health' paradigm[1–7]. Accordingly, there is an increasing interest in exploring environmental and animal-associated microbial ecosystems. At the forefront of the 'One Health' paradigm lies the pressing issue of antimicrobial resistance (AMR)[8]. The emergence and spread of AMR pose a significant threat to public health worldwide, with escalating concerns about its impact on a population scale[9,10]. While extensive literature has documented the prevalence of AMR in farm animals, shedding light on the consequences of intensive animal farming, comparatively scarce data is

[1]Chair for Clinical Bioinformatics, Saarland University, 66123 Saarbrücken, Germany. [2]Institute of Medical Microbiology and Hygiene, 66421 Saarland University, Homburg, Germany. [3]Department of Medicine II, 66421 Saarland University, Homburg, Germany. [4]Helmholtz Institute for Pharmaceutical Research Saarland, Helmholtz Center for Infection Research, 66123 Saarbrücken, Germany. [5]Department of Computer Science, Saarland University, 66123 Saarbrücken, Germany. [6]Zoo Saarbücken, 66121 Saarbrücken, Germany. [7]These authors contributed equally: Georges P. Schmartz, Jacqueline Rehner, Verena Keller, Andreas Keller. ✉e-mail: andreas.keller@ccb.uni-saarland.de

available for wildlife populations[11–15]. However, wildlife animals can travel major distances and interact with other animals through inter-species or intra-species interactions, providing numerous opportunities to acquire and spread AMR along the way[16].

Assessing the genetic makeup of animal-derived microbiomes may not only be useful for quantifying the extent of the AMR crisis but also for searching for potential new antibiotics. Biosynthetic gene clusters (BGCs) found across fungi and bacteria have been discussed as a promising avenue for the discovery of novel antimicrobial compounds[17–21]. These genetic loci encode the machinery for synthesizing bioactive compounds and can be specifically searched for through methods of genome mining in metagenomic data analysis[22–24]. The large biodiversity in both environmental and animal-associated microbiomes sources a plethora of BGCs, of which, unfortunately, only a few have the potential to serve as antimicrobial agents[21,25–27].

A study by Youngblut et al. explored BGCs within the microbiome of various wildlife species, focusing on the diversity and functional breadth of the microbiome while minimizing technical variation across samples[28]. However, challenges faced during sampling can introduce various confounding factors that may significantly disturb down-stream analysis and impact final conclusions. To mitigate some of the previously mentioned challenges, zoo animals, serving as ambassadors for their wild counterparts, yet existing in a controlled environment, have been explored[29–31]. However, most studies working on captive animals focus only on one animal species and often limit themselves to their gut microbiome[32–35].

In this study, we focus on the oral and intestinal microbiota of captive animals, all derived from the same zoo, whose environments are regulated and influenced by the presence of zookeepers and visitors. Our aim is not only to characterize the composition of these microbial communities but also to understand their functional roles, document BGCs, and measure antimicrobial resistance genes. Our investigation extends beyond the confines of captive environments, as we performed a comparative analysis between the microbiomes of captive zoo animals and those of their wild counterparts (Fig. 1a). By scrutinizing the microbiota of these captivated creatures, our study

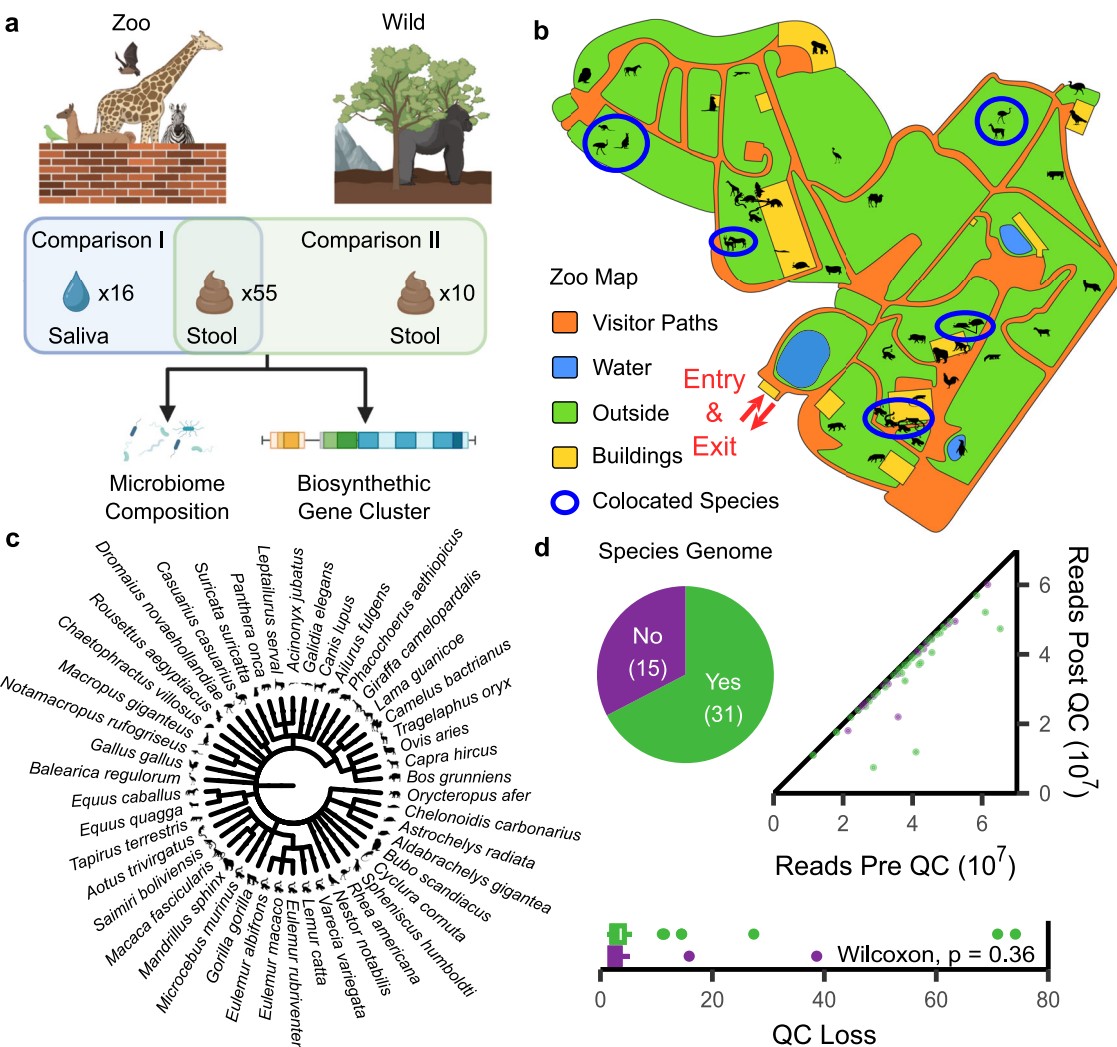

**Fig. 1 | Study setup and data quality. a** The sampling strategy of the study focuses on the comparison of saliva and stool samples of different zoo animals. Extension with the dataset by Youngblut et al.[28] further allows a comparison to wildlife-derived samples. Created with BioRender.com released under a Creative Commons Attribution-NonCommercial-NoDerivs 4.0 International license. **b** Map of the Zoo Saarbrücken with the position of each individual animal species. Co-located animals are encircled in blue. Silhouette-species mappings are elaborated in (**c**). Silhouettes were taken from PhyloPic (phylopic.org). **c** Species included in the study after quality control and introduction of their silhouettes for a large portion of the remaining plots in this study. **d** Statistics on host-derived read decontamination of the metagenomic samples. For datapoints in green, a species-level genome was available to perform read decontamination. Violet datapoints used a taxonomic close substitute genome instead. The p-value indicates the significance of the two-sided Wilcoxon rank sum test on the relative read loss attributed to host contamination (QC quality control).

not only aims to enrich our understanding of the microbiome's complexity but also holds the promise of unearthing novel antimicrobial compounds sourced from animal microbiota.

## Results

### Deep sequencing and quality control results in 64 metagenomes from 45 species

First, we assessed the quality of the metagenomic sequencing results in light of the diversity of species and sample types included and characterized the robustness of our data. We collected a total of 55 stool and 16 saliva samples, representing an extensive range of 48 and 15 distinct zoo animal species (mammals, birds, and reptiles), respectively (Fig. 1b). Subsequently, after sequencing and quality control, we obtained a final dataset comprising 52 stool and 14 saliva samples, reflecting 45 and 13 species (Fig. 1c). Our quality control measures, including host DNA decontamination, yielded minimal read losses during the process, with an average loss of only 6.6% and a standard deviation (SD) of ±13.2%. We retained an average of 5.3 gigabases of sequencing data (SD: 1.7 GB), ensuring a reliable dataset for further analysis.

To account for the species for which a reference assembly was not available on RefSeq, we employed substitute assemblies that were taxonomically close. Notably, this substitution did not significantly impact the relative number of filtered reads (two-sided Wilcoxon $p$-value of 0.36, Supplementary Data 1), supporting our methodology. Utilizing reference-free ordination analysis, we performed an in-depth examination of the cleaned reads, unveiling distinct patterns of sample clustering primarily based on biospecimen (PERMANOVA $p$-value < 0.001, Supplementary Fig. 1). This finding underscores the significance of differentiating between stool and saliva samples and highlights the influence of the animal's specific microbiota on each biospecimen.

### De-novo analysis reveals 585 novel genomes and enhances taxonomic assignment

We encountered an expected—yet significant—challenge when performing taxonomic profiling based on the Genome Taxonomy Database (GTDB) (21). The assignment rate using this database was low, with an average of less than 17% (SD: ±16.6%) matches. This scarcity of read assignments prompted us to adopt a de-novo analysis workflow. Applying this de-novo analysis workflow proved to be instrumental in overcoming some limitations of the taxonomic profiling from existing databases and uncovered the hidden microbial diversity within our dataset. Through this approach, we successfully recovered a total of 786 dereplicated species-level genome bins (SGBs) exceeding the criteria of at least medium MIMAG quality (namely, less than 10% contamination and a minimum of 50% completeness) (22). Among these SGBs, 585 genomes (74%) had no representatives in the GTDB with ANI (Average Nucleotide Identity) less than 95% (Fig. 2a, Supplementary Fig. 2, Supplementary Data 2). Specifically, when examining the stool samples, we found that out of the 616 dereplicated SGBs, 446 had no representatives (72%). In the case of saliva samples, the ratio increased to 139 out of 170 (82%). Saliva samples, accounting for 21% of the overall samples, contributed 22% and 23% of all the dereplicated SGBs and novel dereplicated SGBs, respectively, suggesting the importance of the oral microbiome in uncovering microbial diversity to be on par with the gut microbiome. Analyzing all the recovered SGBs, we observed an average scaffold length of 13 kb (SD: ±2.5 kb). Additionally, we conducted searches for tRNA sequences as well as 5S, 16S, and 23S rRNA sequences within the SGBs. In total, 11,801 tRNAs and 205 rRNAs were detected in the SGBs, averaging 15 tRNAs and 0.3 rRNAs per SGB. Whereas these functional gene statistics are indicative of the overall quality of the assemblies, they also highlight the challenges of reliably assembling ribosomal RNA genes.

Importantly, the integration of our SGBs into the GTDB prior to taxonomic profiling yielded a substantial improvement in the read assignment rate (paired two-sided Wilcoxon $p$-value < $1.7 \times 10^{-12}$, Supplementary Fig. 3). Nevertheless, for 17 samples, the assignment rate remained below the low threshold of 20%. This highlights the significance of including the novel microbial genomes discovered in this study to enhance the accuracy and comprehensiveness of taxonomic assignments. This analysis is also necessary to assess compositional and functional differences between microbiomes and to uncover the distribution of BGCs.

### Culture-based taxonomic assignment yields differences between herbivores and carnivores

In our metagenomic data, the measured alpha diversity, a sign of the microbial complexity of a sample, appears stable for biological replicates. In contrast, the alpha-diversity fluctuates significantly across species (Fig. 3). Astonishingly, we observed a negative Spearman correlation of −0.38 between assignment rates and diversity. Moreover, the reference-based ordination analysis does not yield clear clusters, reflecting neither zoological classification nor diet compositions. Nevertheless, specific zoological proximities are reflected in the clustering hierarchy, such as similar patterns between sheep and goat or between zebra and horse. But in sum, the overall assessment is that the reference-based ordination analysis remains inconclusive with respect to identifying sub-groups of animals. One likely reason for this result is the high variability of assignment rates and missing SGBs. Because differences in the gut microbiota between herbivores and carnivores are known from the literature, we asked whether a more targeted approach involving culturing of bacteria highlights such differences[36,37].

Culturing of 11 saliva and 49 stool samples on TSA, Chocolate blood, Columbia, and MacConkey agar, followed by subsequent MALDI-TOF analysis, enabled the identification of 79 different bacterial species (Supplementary Fig. 4, Supplementary Data 3). While we identified a total of 29 species in saliva (37%), only 8 of them (28%) were also detected in stool samples, where 6 of these 8 were of the genus *Staphylococcus*. In total, 32 species (40%) were only detected in the 38 samples of herbivore animals (including species such as *Enterococcus mundtii, Bacteroides ovatus*, and *Bacillus pumilus*). In contrast, 8 species (10.1%) were observed only in the 7 samples from carnivore animals (including *Citrobacter braakii, Plesiomonas shigelloides,* and *Staphylococcus simulans)*. Moreover, 17 bacterial species (21.3%) were uniquely detected in 15 samples of omnivore animals (including *Neisseria zoodegmatis* and *Staphylococcus hominis* depicting the highest frequency across samples). Across all samples, 7 species (8.8%) are present in all three diet forms, including prevalent intestinal microbiota such as *Enterococcus faecalis, Escherichia coli*, and *Enterococcus faecium*, as well as *Clostridium perfringens* and *Bacillus cereus*. Before adjustment for multiple hypothesis testing, 11 species were significantly unevenly distributed within the cohorts ($\chi^2$ test $p$-value < 0.05). After the Benjamini−Hochberg adjustment, no $p$-value remained significant. Performing the same test over all stool samples and cohorts did not display significant differences between diets ($\chi^2$ test $p$-value = 0.51). As for culturing, only a selection of media was used, bias is introduced by excluding the growth of certain bacteria, that cannot grow on the selected media. However, all samples were treated the same, which makes these results at least comparable. It is worth mentioning that not every microorganism is cultivatable under laboratory conditions, making the metagenomic analysis a more powerful and more precise tool to investigate the microbiome. Nevertheless, the considerably different repertoire of microbiota suggests unique functional characteristics that might be connected to the dietary origin. We thus performed a functional in-silico gene analysis of the respective microbiota.

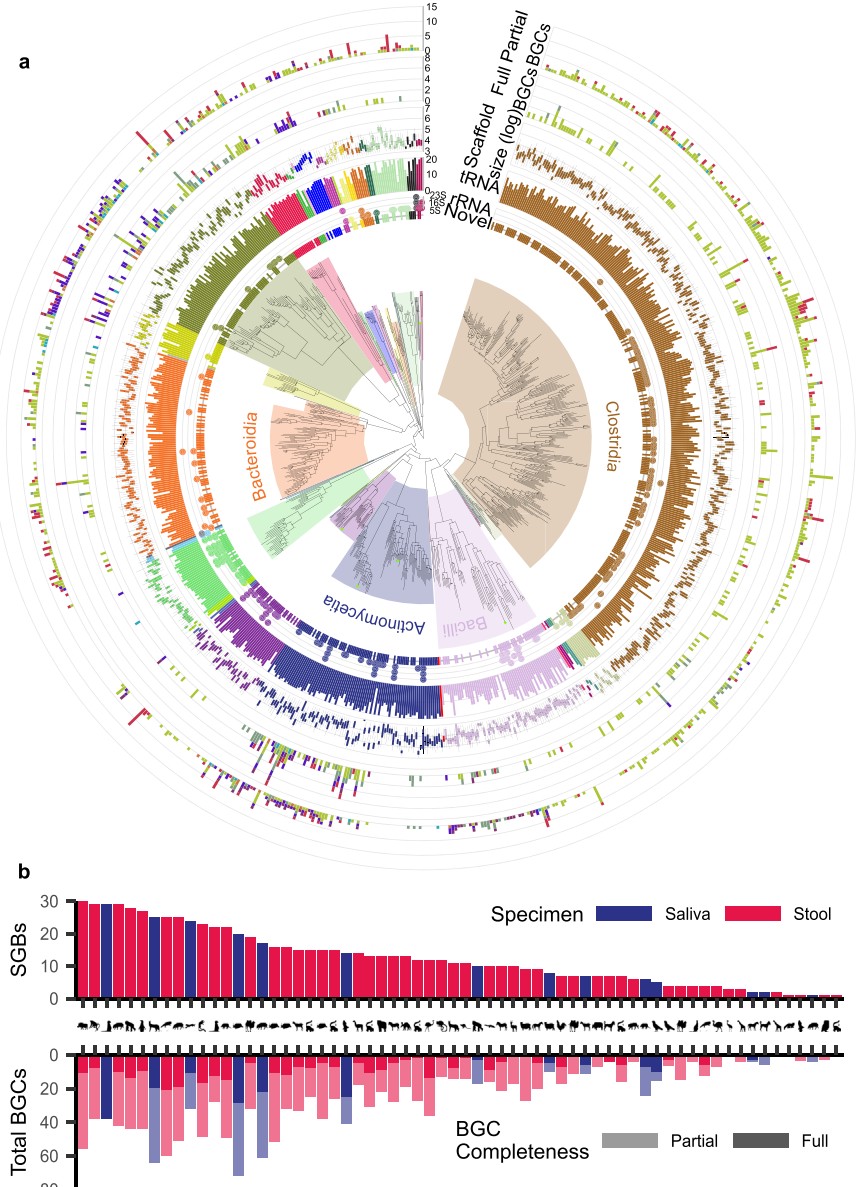

**Fig. 2 | Species-level genome bins. a** Phylogenetic tree of species-level genome bins as classified by the GTDB-Tk. The colored background of clades indicates class ranks. The innermost ring, named *Novel*, indicates if the GTDB-Tk found a species-level assignment. The second, third, and fourth rings discuss bin quality by displaying detected rRNAs, tRNAs, and scaffold length distribution, respectively. The two outer rings indicate the BGCs that were detected in the respective bins. BGCs are classified by type and by completion. A more richly annotated version of this visualization is available in Supplementary Fig. 2. **b** Number of SGBs and BGCs recovered from each sample.

The statistically significant results from this functional gene analysis highlight elevated creatinine degradation I pathway in herbivore animals (Supplementary Fig. 5). Contrastingly, the super pathway of tetrahydrofolate biosynthesis and salvage are more prevalent in microbiota from carnivore animals. Enriched in both carnivore and omnivore animals are bacteria carrying the genomic information for flavin-dependent thymidylate synthase (*thyX*), which is required to synthesize pyrimidine deoxyribonucleotides de novo. Most notably, this gene and the encoded protein are present in human and animal pathogens, such as *Helicobacter pylori*, *Borrelia burgdorferi*, and *Chlamydia trachomatis*[38–40].

### Differences in 484 complete biosynthetic gene clusters depending on the diet

After the initial general functional gene analysis of the different animal microbiota, we looked into the specific metabolite landscapes of individual members of the microbiomes. We performed genome mining of the previously defined SGBs and identified 1588 potential BGCs. Of those, 1104 remained partial, and 484 were identified as full BGC clusters of various categories (Fig. 2a, Fig. 2b, Supplementary Data 4). Further analysis with BiG-SCAPE categorized BGCs into 1482 families, out of which 1407 families were singletons containing only one BGC[41]. A total of five families compromising six BGCs are linked to annotated gene clusters from the MIBiG 3.1 database[42]. But interestingly, BiG-SCAPE did not form any clans of the families. Together with a high number of singleton families, this suggests a high diversity of BGCs in the collected dataset.

With 604 (38%) BGCs, *Clostridia* was the class where we predicted most BGCs. However, this is mostly due to *Clostridia* making up about 36% of our recovered dereplicated SGBs. If we look at the average number of BGCs per SGB and exclude singletons, we observe that, on average, most BGCs were predicted for the class of *Planctomycetia*.

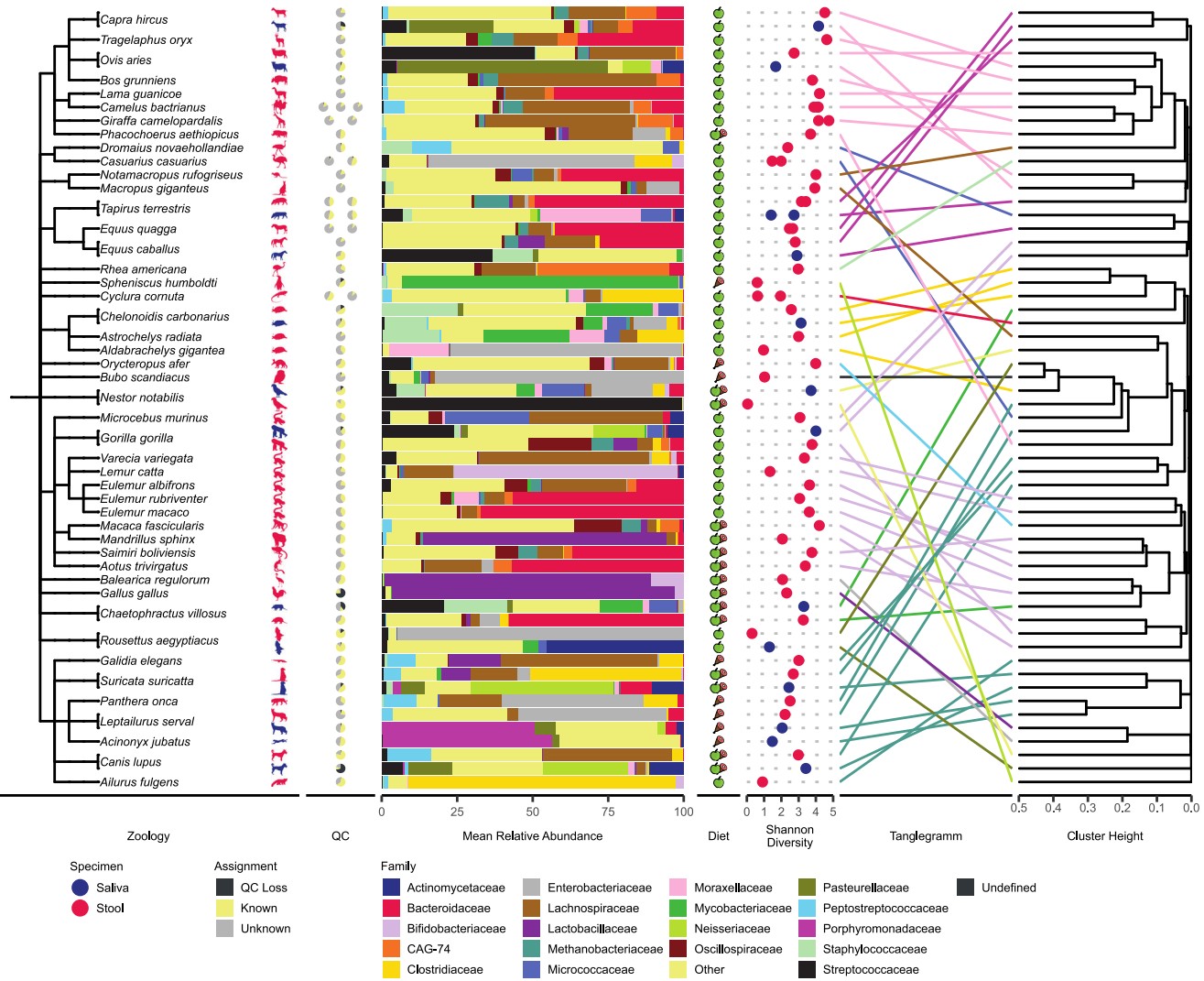

**Fig. 3 | Reference-based analysis.** Summary statistics on quality, diversity, composition, and compositional similarity of microbiomes. Starting from the left, the taxonomic classification of host animals is displayed. Silhouettes represent the host species, and their color represents the different specimens. If multiple replicates were available, multiple pie charts are displayed, where each pie chart indicates the overall quality of the reference-based analysis. Further, diet classification is provided for each species which is consistently used throughout the paper. Three diets are being distinguished: herbivore, carnivore, and omnivore. Alpha-diversity of each sample is indicated using the Shannon index, to visualize microbiome complexity. On the rightmost side, hierarchical clustering based on Bray–Curtis distances are displayed. The optimized tanglegramm displays the accordance between taxonomic class and membership based on predicted microbial composition. The edges are colored by the taxonomic class of the host.

Averaged over 15 genomes, we observed 3.73 BGCs per SGB. With only 4 BGCs in 33 SGBs, *Saccharimonadia* had the lowest non-singleton ratio of BGCs to SGBs. Concerning disparities between the oral and gut microbiome, we observed a total of 450 BGCs (28%) in the 170 saliva-derived SGBs averaging 2.65 BGCs per SGB, which compares to 1,138 BGCs in 616 SGBs at a ratio of 1.85 in the stool samples. Focusing only on the stool-derived BGCs, we observed an average of 2.01, 1.65, and 1.47 BGCs per SGB for herbivores, omnivores, and carnivores, respectively. These differences were confirmed to be significant (Kruskal–Wallis *p*-value < 0.0053). Specifically, the average number of nonribosomal peptides (NRPs) was 2.87 and 3.46 times higher in herbivore SGBs compared to carnivores and omnivores, respectively.

Through a comparative analysis of predicted BGCs and known annotated BGCs from the MIBiG database, we observed 37 BGCs (2%) within our SGBs that shared a similarity of over 50% with known entries. Among these annotations, various compounds may be of relevance to the host organism (Supplementary Fig. 6). We detected virulence factors, such as the toxin tolaasin I, within an SGB derived from tapir saliva. Furthermore, we uncovered various annotations associated with health benefits, including the bacteriocin salivaricin CRL 1328, present in an SGB derived from a mandrill stool sample[43]. We encountered two further compounds with noteworthy properties: α-galactosylceramide, an immunostimulating compound found in an SGB derived from horse stool, and rhizomide, identified in an SGB derived from tapir saliva, exhibiting anti-tumor and antimicrobial properties in vitro[44].

Having captured differences in the repertoire of bacteria from animals with different diets in the gut and oral cavity along with unique functional characteristics and novel BGCs raises the question of whether captivity has an influence on the microbiota or whether wildlife animals reveal similar patterns.

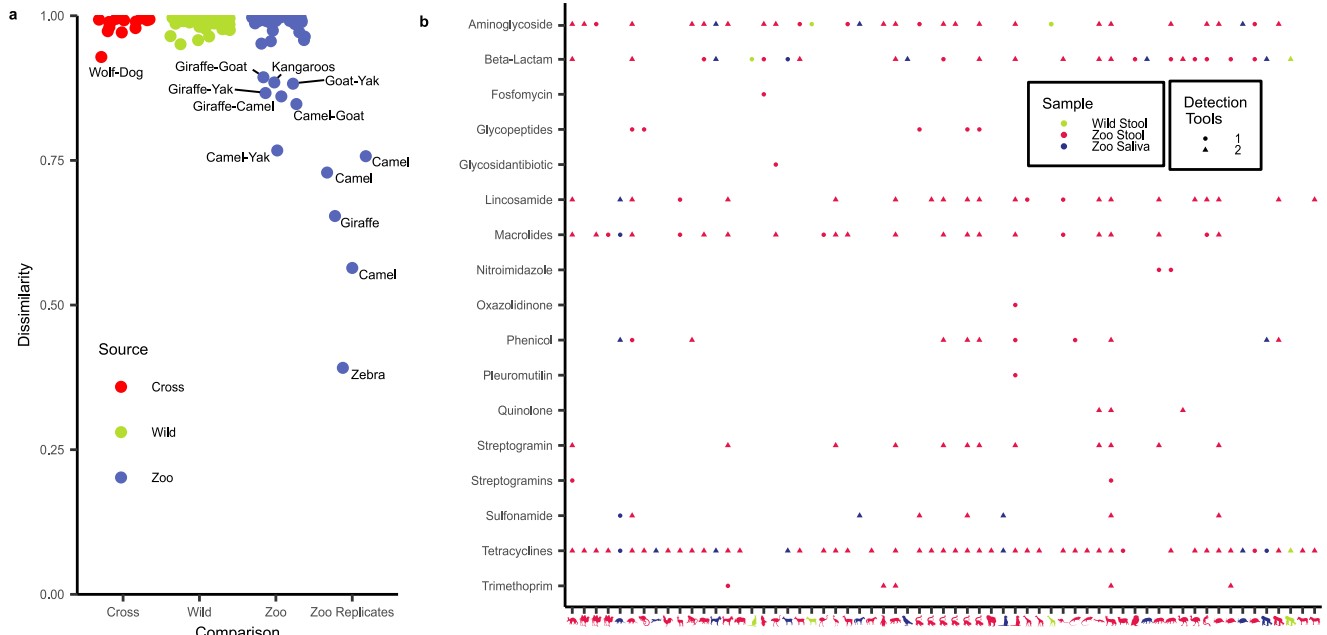

**Fig. 4 | Potential consequences of captivity. a** FracMinHash dissimilarity between samples within our dataset and the dataset of Youngblut et al.[28]. The cross-comparison matches sample pairs as elaborated in Supplementary Data 1. For reference, zoo replicates and their dissimilarity are visualized alongside. **b** Presence of antimicrobial resistance genes for each of the zoo and wildlife samples classified by antimicrobial compound class. Visualized results derive from ResFinder and AMRFinderPlus.

## Animals in captivity present different antimicrobial resistance gene patterns

Comparing microbiome differences between captive and wildlife animals and addressing the complexities in the sample extraction process, we conducted a comparative analysis with data from Youngblut et al.[28], the—as of now—most complete study of animal gut microbiota. Their dataset consisted of 289 samples from 180 different host species, including humans. The large differences between both studies in the selection of animal species call for a balanced and stratified analysis approach. Therefore, we implemented a matching scheme that carefully selects a subset of samples with close zoological similarity from both studies (Supplementary Data 1). We excluded the oral microbiomes of the zoo animals from this analysis because no oral microbiota from wildlife animals were present.

It is important to acknowledge that differences in sample processing between the two studies, such as different DNA extraction kits, can be reflected in the data[45]. Similarly, the easier collection process in a controlled environment, such as a zoo in comparison to a wildlife setting, likely leads to differences in the sample quality. To quantify these differences and ensure methodological consistency, we thus applied our analysis workflow to the selected metagenomes from Youngblut et al.[28]. We observed a significant decrease in read quantity after decontamination compared to the present data, which is explained by the above-mentioned challenges in wildlife sampling (Supplementary Fig. 7a). This reduction also influences the assembly quality, which was lower in the wildlife samples, finally leading to overall shorter fragments (Supplementary Fig. 7b). Consequently, fewer SGBs were recovered in the wildlife samples compared to the zoo dataset (Supplementary Fig. 7c). While the samples from animals kept in captivity retained an average of 9.8 SGBs per sample, the wildlife dataset yielded just 1.9 SGBs. Similar differences also apply to the number and abundance distribution of BGCs. Here, SGBs derived from wildlife animals present, on average, 13 fewer BGCs per SGB (Supplementary Fig. 7d). We were only able to recover partial BGCs in the wildlife samples compared to 50 complete BGCs in the matching zoo samples. Further, only one BGC was annotated to have a similarity

>10% to any known MIBiG BGC. It has a 28% similarity to a carotenoid cluster derived from an *Algoriphagus* species. Again, the latter results might seem counterintuitive, and we might expect more BGCs in wildlife, yet the results are likely biased by the challenges of wildlife sampling and different sample processing. Most importantly, the quality of the wildlife samples is still sufficient to enable reference-free comparison.

As one first aspect, we asked whether the microbiomes between zoo and wildlife animals present a conserved proximity-dependent on the relatedness of host animal species. For the selected samples, we thus performed reference-free FracMinHash comparisons (Fig. 4a). On average, we computed a large dissimilarity between any compared pairs. In detail, the average dissimilarity amounts to 0.98 (SD: 0.018), which is close to the maximal dissimilarity value of 1. Importantly, the dissimilarity distributions within the wildlife and zoo animals do not differ significantly (two-sided Wilcoxon *p*-value > 0.37). Nevertheless, zoo animals display several strong similarities between gut microbiota. These include mostly inter-replicate comparisons of zebra, camel, and giraffes, yielding an overall significantly lower dissimilarity index as compared to the other zoo animals (two-sided Wilcoxon *p*-value < 9.44 × 10$^{-7}$). Of note, no replicates for the wildlife animals are available, explaining the missing similarities within those samples. Interestingly, several of the zoo animal species, including the yak, giraffe, camel, and goat, displayed increased similarities in gut microbiota. The same applies to two kangaroo species that also show similarities in the gut microbiota. Of note, such similarities are not present in the wildlife animals and may suggest an influence, e.g., of the nutrition in this controlled environment. Further, the results clearly argue for combining the advantages of studies in wildlife animals (being closer to nature) and controlled environments (facilitating higher sample quality).

One immediate question in comparing wildlife to captivity setups concerns the presence of AMR. AMR gene analysis of zoo animals revealed potential resistances against antibiotics that are commonly used in veterinary medicine, such as tetracyclines, macrolides, and lincosamides, with *tetW* being the most prominent

gene detected encoding for resistance against tetracyclines, and *lnu(C)* being the most frequent gene encoding for lincosamide resistance when looking at the results using AMRFinderPlus[46] (Fig. 4b, Supplementary Data 5). Genes conferring resistance to fluoroquinolones were rare (*qnrS1*). However, we also observed resistance genes against vancomycin, a broad spectrum antibiotic against Gram-positive bacteria[47]. Specifically, we documented the well-known resistance clusters *vanD* and *vanG*, which are uncommon in humans[48,49]. However, we also detected the *vanO* operon, which has only been identified in captive elephants in Africa[50,51]. Furthermore, we observed a high number of genes encoding beta-lactamases (e.g., blaEC and blaTEM), some of which may even confer resistance to carbapenems. We were able to detect 21 different genes encoding for resistance mechanisms against aminoglycosides, with the most prominent one being *aadE* when looking at the results generated with AMRFinderPlus. We stratified all beta-lactamases according to the Ambler classification and found most prominently beta-lactamases belonging to Ambler class A and C. We did not observe any beta-lactamases belonging to Ambler class B or D. As outlined in the 'One Health' concept, such resistant bacteria could be transferred from zoo animals to zookeepers, increasing the global spreading of such organisms. When we compared our matching stool zoo samples to the wildlife samples, we observed a significantly smaller number of antimicrobial compound classes that are targeted by at least one resistance gene in the wildlife samples (two-sided Wilcoxon *p*-value < 0.036). Overall, we only observed a total of five potential resistances in all analyzed wildlife samples. This suggests that wild animals overall harbour less AMR. Nevertheless, we want to highlight that this result is again to be interpreted in the light of the inferior assembly quality of the wildlife samples, which impacts the quality of AMR gene detection.

## Discussion

Our findings, in line with the study by Youngblut et al.[28], indicate that the microbial dark matter within animal microbiomes remains inadequately characterized in existing data repositories. Despite our extensive efforts and the generation of several novel SGBs, we encountered 17 samples with a low estimated assignment rate below 20%. This deficiency significantly impacts state-of-the-art reference-based analysis, as evident in our own investigation.

The microbial richness we detect, despite the accompanying challenges, presents an intriguing opportunity for the discovery of BGCs associated with antimicrobial natural compounds within these samples. In this context, it is worth emphasizing the advantages of combining different study setups. While our focus lies on samples from a highly controlled environment, specifically a zoo, complementary studies like that of Youngblut et al.[28]. provide valuable insights into wildlife microbiomes, which are closer to the natural microbiota. By integrating findings from diverse settings, we can gain a more comprehensive understanding of the animal microbiome and potentially uncover novel microbial resources with therapeutic potential.

Specifically, the zoo animals present higher numbers of SGBs and BGCs per SGBs but also higher proximity of gut microbiota as compared to the wildlife animals. It is important to acknowledge that the number of BGCs within SGBs can vary, depending on the specific species discovered. However, the improved assembly statistics highlight the advantages of easier sample collection in captivity compared to wild animals, at the cost of BGCs that might only be present in wildlife animals.

When comparing studies, one limitation we encountered was the need to perform inter-species comparisons, which involved species from different continents with potentially diverse diets. This aspect adds complexity to the analysis, as the microbiomes of zoo animals, despite sharing similar diets such as local seasonal vegetables, still exhibit considerable differences. The convergence of microbiome composition across zoo animals appears to be limited, yet measurable.

Furthermore, the presence of AMR genes in animal microbiomes is of considerable importance from the 'One Health' perspective. A previous study, for example, screened captive animals in a zoo in Seoul, South Korea, for particular AMR patterns carried by *Escherichia coli* and *Enterococcus faecalis*. However, the research assessed resistance phenotypically only without analyzing the responsible genes. They found ampicillin resistance in most *E. coli* isolates, and also described multidrug resistance in 50% of isolated *E.coli*[52]. As we wanted to monitor all AMR genes found in captive animals, we followed a metagenomic approach. While it is not uncommon to detect antibiotic resistance genes both in wildlife animals and in zoo animals, the distribution of these genes is of enormous relevance, e.g., to track possible associations between commonly used antibiotics in veterinary and/or human medicine and to decipher potential transmission chains. Also, resistance towards 'last-line antibiotics' in animals might constitute a potential threat to humans[53]. In that regard, it is important to note that we also identified resistance genes against vancomycin in certain animals, including prosimians. Considering their close contact with zookeepers, there is a potential risk of transferring vancomycin-resistant bacteria to humans. A closer look into resistances against beta-lactam antibiotics revealed mainly beta-lactamases belonging to Ambler class A and C, such as *blaEC* and *blaTEM*. Recent findings from studies conducted in Africa revealed mainly *blaOXA, blaKPC, blaNDM, blaSHV*, and *blaVIM* to be found in animals, food, and environmental samples[54]. These genes encoding for carbapenemases were absent in our study but are also becoming a major threat in human medicine[55–57], and hence, are listed among the World Health Organizations's (WHO) 'priority pathogen list in the highest category ' as 'critical'[58]. Therefore, longitudinal screening of captive animals which are in close contact to humans should be employed to notice such a trend before the spread of bacteria carrying such resistances cannot be stopped. The comparatively frequent detection of resistance to tetracycline antibiotics is a further concern and future studies should ideally employ strategies to document the previous antibiotic intake of zoo animals to gain insights on potential associations with a rise in resistance rates. Indeed, as transmission of multi-resistant bacteria has been observed in clinical settings, our findings emphasize the need for comprehensive surveillance and management of AMRs in zoo settings to mitigate potential health risks and maintain a safe environment for both animals and humans[59–61]. Our AMR analysis is limited by the fact that individual resistance genes could not be assigned to specific bacterial species; e.g., *Enterobacter cloacae complex* frequently carries AmpC beta-lactamases in clinical practice, and it would have been interesting to see whether such associations also hold true for captive animals[62].

## Methods
### Study design
For docile animals such as horses, dwarf goats, and tapirs, buccal swabs were easily taken from the oral cavity to collect saliva samples. Concurrently, fresh fecal samples were collected from the enclosures or stables maximum two hours after defecation and immediately transported to the veterinary station. Using a spoon from a stool sample tube, feces from the inner portion of the excreta were transferred into sample tubes. Subsequently, all samples were promptly frozen at −20 °C in the freezer compartment of a refrigerator. Typically, samples were frozen within 30 min of collection.

For non-docile animals, such as primates and large or small carnivores, the same sample collection methods were employed during necessary anesthesia, which occurred for veterinary examinations, treatment, transport, or sex determination. For small animals, fecal samples were collected rectally as swabs, following the same protocol described above, and stored frozen until further analysis. Due to the non-invasive sampling procedure, no ethical approval was required.

## DNA extraction

We extracted whole-genome DNA from all fecal and salivary swabs using the Qiagen QiAamp Microbiome Kit (Qiagen, Hilden, Germany)[8]. The DNA extraction procedure was conducted according to the manufacturer's protocol. Briefly, all swabs containing native samples were vortexed in 1 ml PBS for 2 minutes. The PBS containing the microbes from each sample was then used for DNA extraction according to the manufacturer's recommendation. We used the MP Biomedicals™ FastPrep-24™ 5 G Instrument (FisherScientific GmbH, Schwerte, Germany) for mechanical lysis of bacterial cells. The velocity and duration were adjusted to the 'hard-to-lyse' protocol, meaning 6.5 m/s for 45 s 2 times and 5 min storage on ice in between each lysis step. DNA was eluted in 50 µl elution buffer. The DNA concentration after elution was determined via NanoDrop 2000/2000c (ThermoFisher Scientific, Wilmington, DE) full-spectrum microvolume UV–Vis measurements[45].

## Library preparation and sequencing

Extracted whole-genome DNA was sent to Novogene Company Limited (Cambridge, UK) for library preparation and sequencing. For quality control of the samples, potential genomic DNA degradation was measured using the fragment analyzer platform AATI (Agilent Technolgies, CA, USA). The DNA concentration was measured using Qubit (Thermo Fisher, Wilmington, DE) before library preparation. Briefly, samples were subjected to metagenomic library preparation and further sequenced via paired-end Illumina Novaseq X plus Sequencing PE150. For library preparation, the Novogene NGS DNA Library Prep Set (Cat No.PT004) was used. Genomic DNA was sheared into short fragments in random positions and fragmented DNA was subjected to end-repair and A-tailing, as well as Illumina adapter ligation. Fragments with the appropriate size of 500 bp were selected via beads-based size selection of libraries and amplified via PCR. The PCR products underwent quality control and quantification using the Qubit system and bioanalyzer to visualize the generated fragment sizes. All samples were pooled and sequenced on the Illumina Novaseq X plus Sequencer. For all samples, 5 Gb reads per sample were generated.

## Culturing of bacteria

Native fecal samples were streaked out using the swab they were taken with, on three different agar plates: TSA with 5% sheep blood (TSA), MacConkey (MC), and Columbia (Co) agar plates (Becton, Dickinson and Company, Heidelberg, Germany). Oral samples were streaked out on TSA, Co and Chocolate blood (CB) agar plates (Becton, Dickinson and Company, Heidelberg, Germany). All TSA, CB, and MC agar plates were incubated at 35.6 °C and 5% $CO_2$ for a minimum of 18 h and a maximum of 24 h. Co agar plates were used for the cultivation of anaerobic bacteria and therefore incubated in an anaerobic environment for a minimum of 48 h at 35.6 °C[45].

## Mass spectrometry-based identification

Bacterial colonies obtained by culturing native fecal and oral samples on different agar plates were subjected to species identification using matrix-assisted laser desorption/ionization time-of-flight (MALDI-TOF) mass spectrometry. To this end, colonies from overnight growth on tryptic soy agar plates containing 5% sheep blood, from Columbia agar plates, and MacConkey agar plates were taken with a sterile toothpick and spotted on the MALDI-TOF target plate by smearing one colony on one spot of the target, dried, and then overlayed with 1 µl of 70% formic acid. This step aids in the cell lysis and makes peptides and proteins available for ionization. After drying, 1 µl of α-cyano-4-hydroxycinnamic acid (CHCA) matrix solution (Bruker Daltonics, Bremen, Germany) was pipetted on top of the bacterial matter and formic acid and set to dry. The matrix solution is composed of saturated CHCA dissolved in 50% (v/v) acetonitrile, 47.5% (v/v) LC-MS grade water, and 2.5% (v/v) trifluoroacetic acid. After drying the matrix solution at room temperature, each spot was overlayed with 70 % formic acid to pre-disrupt the cells. Followed by drying at room temperature, the plate was placed into the Microflex LT Mass Spectrometer (Bruker Daltonics) for MALDI-TOF MS. All measurements were performed with the AutoXecute algorithm in the FlexControl© software version 3.4 (Bruker Daltonics). Each spot was excited with 240 laser shots in six random positions. Measurements were carried out automatically to generate protein mass profiles in linear positive ion mode using a laser frequency of 60 Hz, high voltage of 20 kV, and pulsed ion extraction of 180 ns. Mass charge ratio ranges (m/z) were measured between 2 kDa and 20 kDa. We identified bacterial species using the software MALDI BioTyper compass explorer (v.3.0). The database used was Bruker´s commercial database: Bruker BDAL database (10,148 species-specific main spectra profiles). Identification scores above 2.0 were considered a precise identification of proteins and peptides on the species level, scores between 1.7 and 1.99 were considered as possible species identification and precise genus identification, and all identification scores below 1.7 were considered unsuccessful identification. In this study, we only considered scores ≥2 for analyses[36].

## Next-generation sequencing preprocessing

The first step of data analysis was host read removal with KneadData (version (v):0.10.0; command line arguments (cla): "--trimmomatic-options = 'LEADING:3 TRAILING:3 MINLEN:50' --bowtie2-options = '--very-sensitive --no-discordant -reorder'") using the respective genomes as specified in Supplementary Data 1[63]. The selected, publicly available, host genomes were downloaded with the ncbi-datasets-cli (v13.35.0). For several animal species, no exact sequenced genome of sufficient quality was available and instead, a taxonomically close substitute was selected. Bowtie2 (v2.4.5; -s) databases were prepared for each reference[64]. After decontamination, we performed sequence overrepresentation analysis and quality assurance with fastp (v:0.23.2; cla: --overrepresentation_analysis) and visualized results with MultiQC (v1.13a)[65,66]. The two-sided Wilcoxon rank sum test was performed on the relative loss attributed to host DNA removal. To reduce bias, replicates were averaged. Saliva and stool samples were not averaged.

## Metagenome assembly

We assembled each sample with SPAades (v3.15.4; cla: --meta) and monitored assembly quality with QUAST (v5.0.2; cla: -s)[67,68]. Next, we aligned each host decontaminated sample against each set of assembled scaffolds with BWA-MEM2 (v2.2.1) and generated abundance profiles for each combination[69]. We extracted coverage information to bin scaffolds with MetaBAT2 (v2.15; cla:l --seed 420 --unbinned), MaxBin2 (v2.2.7), and DAS Tool(v1.1.5; --search_engine diamond)[70–72]. MAGs across all samples were aggregated and dereplicated with dRep (v:3.4.0; cla: -comp 50 -con 10 --checkM_method lineage_wf --S_algorithm fastANI --S_ani 0.95 -nc 0.5). At last, we used GTDB-Tk (v:2.1.1; cla: classify_wf), tRNAscanSE (v:2.0.11;--brief -Q), and barrnap(v:0.9; cla: -q) to taxonomically classify MAGs and annotate them with tRNA and rRNA information based on their classified kingdom[73,74].

## Reference-based compositional analysis

FracMinHash profiles were computed for all samples with sourmash (v:4.4.3; cla: -k51)[75]. After FracMinHash profile generation, samples were compared with sourmash compare. Dissimilarities were computed by subtracting the resulting similarities from one. Samples were embedded with UMAP (v:0.2.8)[76]. Further, for each SGB, FracMinHash profiles were computed as well, and an index was generated. The PERMANOVA analysis treated samples and replicates as independent[77]. Taxonmic profiling was performed with sourmash (cla: -k51) our previously generated indices, GTDB (v:GTDB R07-RS207 all genomes k51), and host decontaminated reads. Shannon index was used as the alpha-diversity measure and computed with phyloseq (v:1.40.0)[78,79]. Relative abundances were averaged if replicates were available. Clustering was

performed with average hierarchical clustering on Bray-Curtis distances computed with the vegan package on mean relative abundances (v:2.6.2)[80]. Tanglegram was optimized for visual clarity with "step2side" algorithm of the R dendextend package (v:1.16.0)[81]. Differential abundance analysis was made with ANCOMBC (v:1.6.2) comparing herbivores and the union of omnivores and carnivores[82].

## Functional analysis

In order to incorporate our own SGBs into the functional profiling step, we updated an existing GTDB207-based database with Struo2 (v:2.3.0)[83]. After database generation, functional profiling was performed with HUMAnN 3 (v:3.6; cla: --bypass-nucleotide-index)[63]. We also used ANCOMBC for exploration of differences in function. The default setting of Holm–Bonferroni *p*-value adjustment was employed. Genes were predicted with prodigal (cla:-p meta) and passed to antiSMASH (v:6.1.1; cla: -cb-knownclusters --cb-subclusters --asf) for BGC detection[22,84]. A BGC was classified as partial if it is shorter than 5 kbp or located on a contig edge and as full otherwise. Clustering of all BGCs was performed with BiG-SCAPE (v:1.1.5; cla: --mibig) using Pfam (v:35.0). BiG-SCAPE failed to process two BGCs and removed them from further analysis[42,85].

## Antimicrobial resistance gene analysis

Antimicrobial resistance gene assessment was performed with AMRFinderPlus (v:3.11.26; database v:2023-11-15.1; cla: --report_all_equal --plus --coverage_min 0.9 --ident_min 0.95), DeepARG (v:1.0.4; database v:2; cla: --model SS --type nucl −min-prob 0.8 --arg-alignment-identity 95 --arg-alignment-evalue 1e-10 --arg-num-alignments-per-entry 1000), and ResFinder (v:4.4.2; database v:2.3.0; clr: --threshold 0.95 --min_cov 0.95). After metagenomic assembly, the contigs of each sample were passed to each of the aforementioned tools, grouping contigs by sample. Results across all samples, as well as tools were aggregated using hamronizer (https://github.com/pha4ge/hAMRonization; v:1.1.4)[86–88]. DeepARG predictions were rejected from further analysis due to exceptionally high numbers of detected resistance genes and divergence from manually inspected results. AMRFinderPlus predictions were discussed in detail in the paper. For visualization, ResFinder and AMRFinderPlus results were unified if genes that provide resistance against the same group of antimicrobial compounds were predicted on the same contig within a 50 bp interval by both tools.

## Wildlife comparison

Samples specified in Supplementary Data 1 were downloaded from the European Nucleotide Archive and processed identically to our dataset, from host DNA removal to BGC prediction[89]. We subsetted our data to only the paired samples specified in the aforementioned table. Pairings were manually selected based on taxonomic similarity. Paired comparison to our data was done based on FracMinHash dissimilarities.

## Reporting summary

Further information on research design is available in the Nature Portfolio Reporting Summary linked to this article.

# Data availability

The raw unfiltered sequencing reads as well as dereplicated SGBs generated in this study have been deposited in the Sequence Read Archive under the accession PRJNA983076.

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

## Acknowledgements

The compute infrastructure for this project was funded by the DFG [469073465], received by AK. This work was supported financially by Saarland University, the UdS-HIPS TANDEM initiative and the TALENTS Marie Skłodowska-Curie COFUND-Action of the European Union [101081463], received by LAGM. The views and opinions expressed are, however, those of the authors only and do not necessarily reflect those of the European Union, which cannot be held responsible for them.

## Author contributions

A.K., V.K., and R.F. had the idea for this study. G.P.S., J.R., S.L.B., and A.K. developed the methodology. R.F. collected samples and managed the animals. J.R. and M.J.S. performed whole-genome DNA extraction, culturing, and MALDI-TOF analysis. B.G.C. interpretation was performed by G.P.S., A.G., and R.M. S.L.B., V.K., R.M., M.K., and A.K. provided financial resources and supervised the project. J.R. supervised experimental investigations and project coordination. G.P.S., A.T., L.A.G.M., and A.K. performed computational analysis and data processing. G.P.S. and J.R. drafted the paper. All authors have read and critically reviewed the paper, in particular intellectual content, and agreed to the submission of the paper.

## Funding

## Competing interests

G.P.S., R.M., and A.K. are co-founders of MooH GmbH, a company developing metagenomic-based oral health tests. The remaining authors declare no competing interests.
