## [Peer Review File · Nature Communications]

Exploring Microbial Diversity and Biosynthetic Potential in Zoo and Wildlife Animal MicrobiomesReviewers' Comments:

Reviewer #1:

Remarks to the Author:

This study examines variation in microbiota (gut and oral) and antimicrobial resistance genes between captive and wild animals. While the premise of the study is certainly interesting and valuable, there are a number of concerns that warrant serious attention. The below feedback is not a comprehensive review but is meant to identify the major flaws and provide associated feedback to the authors.

First and foremost, the complexity of studying this many animal species is largely overlooked in this paper. While it is understandable that the authors are attempting to wrangle a very complex dataset into a manageable analysis, there needs to be greater attention given to the significant ecological and phylogenetic variation between animal individuals and species. For example, the authors should consider applying phylogenetic comparative approaches and/or assess phyllosymbiosis to better elucidate the underlying structure of microbiota variation across host taxa. This may be particularly relevant when comparing captive and wild species as there have been studies that demonstrate differential signals of phylogenetic patterns in wild animals vs captive animals. In addition, the authors should be very wary of ascribing robust meaning to only one sample per species per environment. There is significant variation across individuals of a species, even within identical environments. In other words, while there are an impressive number of species in this study, there is still only an $n=1$ within each unit of analyses (a species within a captive or wild environment).

With AMR being so prominent in the title of the paper, it was surprising to see very few results on AMR genes. There is little detail outside of vancomycin resistance genes, which while interesting, are only one of many classes of clinically-relevant AMR genes. If this is a significant 'storyline' of the paper, there needs to be greater spaces dedicated to detailing these results. There is also not a single mention of AMR or antimicrobial resistance in the methods. It is also extremely odd – bordering on unbelievable – that there were only 4 AMR genes identified in all of the wildlife samples. This conflicts with a multitude of existing papers showing high diversity and abundance of AMR genes in wildlife, including clinically relevant genes. Although it is intuitive that wild animal would harbor less diverse and lower abundance AMR profiles compared to captive animals, only finding 4 AMR genes across all the wildlife microbiomes is a definitely outlier in the context of the larger literature. The authors should delve deeper into this and seriously consider whether this is an artifact or a true biological pattern. Without clear methods on how the AMR genes were identified and quantified, these issues cannot be robustly reviewed.

The authors should be aware and transparent that, while the data processing was identical between wild and captive samples, it seems unlikely (and is not specified) that the sample collection & processing, DNA extraction & preparation, and sequencing parameters used for wild samples were the same as those for the captive animal samples. Different extraction kits, in particular, are known to skew microbial results. This may be an unavoidable component of 'meta-analysis' studies such as this, but these limitations need to be clarified up front.

In addition, there are some gaps in the methods. There is no specification of how fresh the stool samples were (what was the maximum time between defecation and collection?). There is also no discussion that some of the “stool” samples are actually rectal swabs (which have been shown to vary in their value as a proxy for fecal microbiota across different species). The library Preparation and Sequencing methods are practically non-existent. The authors state that, for species that did not have a reference genome, a “taxonomically close substitute” was used. But those substitute species are not listed anywhere I could find.

Finally, this paper is difficult to read and internalize, mainly because the goals of the study are rather unclear. The Introduction starts with a focus on humans (which are not included in the study) and then goes on to focus on One Health, but is actually only studying non-human animals. Animals are certainly an important component of One Health but the Introduction lacks crucial background on existing research into animal microbiota and AMR profiles. The discussion then starts with the idea of ‘microbial dark matter’ which isn’t really mentioned in the Introduction. Then, the Discussion is just an extremely brief summary of the results, with a total of 4 citations (!). There needs to be significantly more discussion of the biological relevance of these results and their context within the larger literature. It is unclear whether this is an issue of word limits but regardless, the current Discussion is insufficient. Overall, I would strongly urge the authors to carefully consider their objectives for this manuscript and better highlight the strengths of this studying while also acknowledging its limitations.

Reviewer #2:

Remarks to the Author:

In this study, the oral and intestinal microbiomes of captive wildlife species across various taxa were investigated. The aim was to characterize the microbiota composition, metabolic pathways, AMR genes, and biosynthetic gene clusters (BGCs) in about 45 wildlife species sampled in a zoo environment. The data derived from captive wildlife species were compared to published data from free-ranging wildlife species.

The study accumulated interesting data that might provide the baseline for many important research questions. It assembled a large amount of species-level genome bins (SGBs), predicted complete BGCs, reported that the diet influence variations in metabolic pathways, as well as differences in the microbiome between captive and free-ranging wildlife. The species match was taxonomically as close as possible but not a pairwise comparison per species was feasible. Moreover, and I think most interestingly, the work unveiled AMR genes against common veterinary antibiotics and resistance to vancomycin, a critical antibiotic in human medicine.

Though the study has many merits, and a large amount of lab and bioinformatic work has been invested, the results section lacks a clear flow of in-depth analyses in relation to clearly defined research questions. Clearly developed hypotheses and predictions in relation to already available knowledge are missing. It is not surprising (and new) that the microbiomes differ according to diet

(herbivores-omnivores-carnivores), phylogeny, stool – saliva, and also between captive and free-ranging wildlife. But I am sure more elaborated questions could be asked.

Very interesting are the findings about AMR in zoo animals to a common antibiotic in humans. Specifically, they observed resistance genes against vancomycin, which is a last resort antibiotic against infections with Gram-positive bacteria in human medicine.

I think the data are not sufficiently explored and might hide a wealth of information that could be investigated in more detail. I appreciate the data outline and indication of data quality in Fig. 1. Looking at the study design, with the (partly) unbalanced sample, methodological challenges and data output I suggest to develop more specific research questions. An important question could be to follow up the AMR observations. E.g., understanding the spread of AMR between species and humans according to contact probability and horizontal gene transfer. It is not only a zoo-relevant question to investigate advantages/disadvantages of animal co-housing and contact to humans on microbiota communities and health, in generating an increased or decreased microbial diversity, and especially the impact of horizontal gene transfer (\diamond AMR). Along this line, the Intro is quite broad and unfocused and should be streamlined.

Reviewer #3:

Remarks to the Author:

The manuscript deals with the microbiomes of several wild and captive animal species and compares also the differences between wild and captive for certain species. It is most interesting from a descriptive point of view, and less from an analytical point of view comparing the different animal species as for most species only one sample was included.

While the introduction is focused on the differences in microbiomes by diet,... (line 41), however it could include also the functional profiles which in general differ less.

It is a pity that for most animal species included only one sample is available and as such the intraspecies variation cannot be taken into account.

The inclusion of culture-based determination of the microbiota is certainly a addition to the knowledge of the microbial diversity. The comparison between saliva and stool samples is however not relevant as these are quite different microbiota. The description of that part is however quite limited and mainly included in the supplemental material. The use of mostly non-selective plates has probably also lead to the limited number of different species isolated. It would have been good to include a larger diversity of plates for the isolation of species seen this is compared to the non-culture based methods. The comparison between herbivores, carnivores and omnivores is also not very relevant for the culture-based methods. Moreover, there may be differences between the age within an animal species. This is for example very well known for poultry.

The statement on line 157 that only 30-60% is cultivable needs referencing. According to my information it is much less.

The part on antimicrobial resistance is not clear. It is not clear what is detected in general and how the analysis was made, was the focus on acquired resistance genes only or has a more general

database as CARD being used of the assessment? Different databases may give quite different results. Showing the data as resistant against a certain antimicrobial is not that helpful in understanding the data. It is unclear why the specific focus is on vancomycin resistance. VanO has moreover been described in elephants and thus has been detected before in animal samples. The part on the transfer of resistance within the manuscript as this needs a very different methodology to assess. It should be noted that in all ecosystems antimicrobial resistance genes are found.

Reviewer #1 (Remarks to the Author):

This study examines variation in microbiota (gut and oral) and antimicrobial resistance genes between captive and wild animals. While the premise of the study is certainly interesting and valuable, there are a number of concerns that warrant serious attention. The below feedback is not a comprehensive review but is meant to identify the major flaws and provide associated feedback to the authors.

We appreciate that the reviewer made efforts to highlight the most significant challenges with our study here. With the comments below we address those but remain of course open for more specific feedback.

First and foremost, the complexity of studying this many animal species is largely overlooked in this paper. While it is understandable that the authors are attempting to wrangle a very complex dataset into a manageable analysis, there needs to be greater attention given to the significant ecological and phylogenetic variation between animal individuals and species. For example, the authors should consider applying phylogenetic comparative approaches and/or assess phyllosymbiosis to better elucidate the underlying structure of microbiota variation across host taxa. This may be particularly relevant when comparing captive and wild species as there have been studies that demonstrate differential signals of phylogenetic patterns in wild animals vs captive animals. In addition, the authors should be very wary of ascribing robust meaning to only one sample per species per environment. There is significant variation across individuals of a species, even within identical environments. In other words, while there are an impressive number of species in this study, there is still only an $n=1$ within each unit of analyses (a species within a captive or wild environment).

Many thanks for the comprehensive assessment. From the comments above we extracted three separate points that we split below in order to improve the clarity of our reply:

- 1. **Replicates and Individual Variation:** We acknowledge the limitation of having only one sample per species per biospecimen. To mitigate this, we've taken a cautious approach in our analyses by grouping species when making comparisons. As examples, we provide the antimicrobial resistance comparison to wildlife samples where comparisons were made across several species, and comparisons between herbivores and carnivores, where cohorts were constructed. We've also refrained from drawing definitive conclusions based solely on individual species or samples, recognizing the inherent variability within species and environments. Last but not least, we mention this aspect even more explicitly in our revised version. Extending the study to multiple zoos is certainly an aspect we will address in the future (and initiated already discussions) but this significant endeavor is beyond the scope of what we can provide in the revision in a reasonable time*
- 2. **Ecological Variation Analysis:** Our manuscript employs state-of-the-art ecological ordination analysis to explore ecological variation between animal species. Additionally, we've supplemented this with MinHash-based dissimilarity analysis to account for lower assignment rates in some samples. This MinHash-based analysis is agnostic to any reference database or taxonomic profiling method as conclusions are drawn from k -mers computed on NGS reads. While we understand that both these methods may have limitations, we believe they provide valuable insights into ecological variation and adequately address the challenges of our data.*
- 3. **Phylogenetic Comparisons:** We agree with the reviewer that adequate analysis of this dataset requires phylogenetic comparative approaches. Especially the assessment of phyllosymbiosis is of special interest. To this end, we closely followed the proposed workflow by Lim and Bordenstein elegantly summarized in Figure 3 of their respective manuscript (1). The main discrepancies in our workflow used to generate our third figure and their proposed workflow are found in the description of the host species. Instead of constructing on the left-hand side of the plot a phylogenetic tree for the host species, we instead went with the zoological classifications. However, we argue that zoological classification is close enough to the*

phylogenetic tree to not distort general conclusions drawn from our analysis. Concerning the comment tailored on wildlife animals, we refrained from including the same wildlife samples in this analysis for several reasons. First, the proposed methodology relies heavily on the abundance information during tree generation. The counts, however, are heavily biased as we only deduced SGBs for zoo samples introducing a confounding factor for the meta-analysis. Further, we wanted to avoid even more data overload. Nevertheless, we performed the suggested analyses and provided them in Reviewer Figure 1: Phyllosymbiotic comparison of samples. We repeated the taxonomic profiling using our constructed reference material. Clustering was performed on Bray-Curtis distances using average hierarchical clustering:

Reviewer Figure 1: Phyllosymbiotic comparison of samples. Blue names represent zoo animal samples collected throughout this study. Samples of wildlife animals are written in green font.

With AMR being so prominent in the title of the paper, it was surprising to see very few results on AMR genes. There is little detail outside of vancomycin resistance genes, which while interesting, are only one of many classes of clinically-relevant AMR genes. If this is a significant ‘storyline’ of the paper, there needs to be greater spaces dedicated to detailing these results. There is also not a single mention of AMR or antimicrobial resistance in the methods. It is also extremely odd – bordering on unbelievable – that there were only 4 AMR genes identified in all of the wildlife samples. In This conflicts with a multitude of existing papers showing high diversity and abundance of AMR genes in wildlife, including clinically relevant genes. Although it is intuitive that wild animal would harbor less diverse and lower abundance AMR profiles compared to captive animals, only finding 4 AMR genes across all the wildlife microbiomes is a definitely outlier in the context of the larger literature. The authors should delve deeper into this and seriously consider whether this is an artifact or a true biological pattern. Without clear methods on how the AMR genes were identified and quantified, these issues cannot be robustly reviewed.

The reviewer mentions two important points. First, as also mentioned by the other reviewers, we partially missed to have a consistent storyline. This is explained in the genesis of the manuscript

partially and we corrected it over the revision. The title is only one of the respective issues. We fully agree that the AMR aspect is far too prominent in the title, because it in fact is only a minor aspect of the final manuscript, especially after the revision. The new title reads: "Exploring Microbial Diversity and Biosynthetic Potential in Zoo and Wildlife Animal Microbiomes".

The second point is scientifically even more important. We indeed claimed originally a limited number of resistances in wildlife. This is due to two factors: the first one is the usage of conservative measures for resistance and the second one is the technical challenges with the samples from the wildlife study. In the revised manuscript we provide a deeper analysis of the resistances in caged and wildlife animals, specifically, we used three different bioinformatic tools to detect AMR genes from metagenome assembled NGS data. The results of AMRFinderPlus, DeepARG, and Resfinder were unified using hAMRonizer (<https://github.com/pha4qe/hAMRonization>) and discussed anew within the manuscript (2-4). The adapted method section can be found in line 467 ff. We hope that this consensus approach is to the satisfaction of the reviewer.

The authors should be aware and transparent that, while the data processing was identical between wild and captive samples, it seems unlikely (and is not specified) that the sample collection & processing, DNA extraction & preparation, and sequencing parameters used for wild samples were the same as those for the captive animal samples. Different extraction kits, in particular, are known to skew microbial results. This may be an unavoidable component of 'meta-analysis' studies such as this, but these limitations need to be clarified up front.

We apologize for having raised the impression of not being transparent in this crucial aspect. In fact, this point was one of the dominating ones in our analysis. It is endeavored to foster transparent communication by conducting thorough analysis and employing data visualization on this topic, as e.g. outlined in line 248 of the original manuscript. Our analysis revealed significant disparities in the overall quality of the data, which manifested in variations in assembly length, among other factors. However, we regrettably overlooked mentioning that differences in sample preparation could also contribute to these quality discrepancies. Instead, we solely attributed them to the sampling process, thereby neglecting to mention many other potential confounding factors. Indeed, we have even carried out an own original study to address the point of impact of sample extraction kits that we missed citing in the original manuscript ("Systematic Cross-biospecimen Evaluation of DNA Extraction Kits for Long- and Short-read Multi-metagenomic Sequencing Studies") (5). In our revised manuscript, indicated by the forthcoming modifications in lines 226, we aim to rectify this oversight. We express our gratitude to the reviewer for highlighting this point.

As one last consequence of the previous point, this comment and comments of reviewer 2 we make the comparison to wildlife and caged animals only one aspect while focusing more on our original goal: understanding whether zoo animals still have a diverse microbiome despite a rather standardized nutrition and care.

In addition, there are some gaps in the methods. There is no specification of how fresh the stool samples were (what was the maximum time between defecation and collection?). There is also no discussion that some of the "stool" samples are actually rectal swabs (which have been shown to vary in their value as a proxy for fecal microbiota across different species). The library Preparation and Sequencing methods are practically non-existent. The authors state that, for species that did not have a reference genome, a "taxonomically close substitute" was used. But those substitute species are not listed anywhere I could find.

*We significantly improved the methods section. For example, we specify that the maximal time between defecation and collection in the zoo was below 2 hours. We also explicitly list the information on the rectal swabs in the Supplementary data Table 1. We also expanded the information on library preparation and sequencing, which can be found in the methods section. The mentioned "taxonomically close substitutes", can be found in the same table, and are listed as *reference* "genome*

used for decontamination” in the manuscript. Lastly, for the difference in microbiome between rectal swabs and stool samples, we agree with the reviewer that differences in sampling strategy are likely going to reflect in the measured microbiome as has been highlighted in the literature. As the wildlife material has been stool we aimed to sample from the same specimen type. However, as both, the safety of the animals as well as the zoo staff have been our highest priority for the entirety of this study, we believe that the rectal swabs were a necessary compromise for some of the more dangerous animals to interact with. Nevertheless, we agree with the reviewer that this shortcoming should be highlighted more explicitly in the discussion section of the manuscript. Hopefully of interest to the reviewer, we recreated Supplementary Figure 1 with only stool/swab samples and coloured them to provide an estimate of the overall effect of different sampling strategies in this study (Reviewer Figure 2).

Reviewer Figure 2: Minhash embedding of zoo samples colored by detailed sampling strategy. Oral samples were excluded.

Finally, this paper is difficult to read and internalize, mainly because the goals of the study are rather unclear. The Introduction starts with a focus on humans (which are not included in the study) and then goes on to focus on One Health, but is actually only studying non-human animals. Animals are certainly an important component of One Health but the Introduction lacks crucial background on existing research into animal microbiota and AMR profiles. The discussion then starts with the idea of ‘microbial dark matter’ which isn’t really mentioned in the Introduction. Then, the Discussion is just an extremely brief summary of the results, with a total of 4 citations (!). There needs to be significantly more discussion of the biological relevance of these results and their context within the larger literature. It is unclear whether this is an issue of word limits but regardless, the current Discussion is insufficient. Overall, I would strongly urge the authors to carefully consider their objectives for this manuscript and better highlight the strengths of this studying while also acknowledging its limitations.

Our original goal of the study was to examine whether zoo animals have a diverse microbiome or a more limited one, e.g. because they get a much more uniform nutrition than in wildlife. In the revised manuscript we clearly emphasize the aspect. In this regard:

- *Introduction:*
 - *We removed parts of the introduction on humans but feel that they still should be mentioned*
 - *We added much more literature on animals, especially but not limited to AMRs*

- *We clearly mention the main hypothesis and reason for the study: understanding microbial compositions in zoo animals*
- *Discussion:*
 - *We better mention limitations*
 - *We put our results in the context of other studies. While we feel that the number of references is not the best measure we fully agree with the reviewer that mentioning only 4 manuscripts is certainly a shortfall and have now 11 manuscripts cited.*
 - *As mentioned in the general letter to the editor above, we focus more on our strengths, namely a high-quality collection of metagenomes in many species in a very controlled environment but clearly (!) mention the limitations that this reviewer pointed at.*

In summary to reviewer 1's comments, we highly appreciate this critical but nonetheless constructive comment. We hope that the reviewer now feels more comfortable with the study, the storyline, and main objectives and how we put them into context. As said, we are open for other more specific comments.

Reviewer #2 (Remarks to the Author):

In this study, the oral and intestinal microbiomes of captive wildlife species across various taxa were investigated. The aim was to characterize the microbiota composition, metabolic pathways, AMR genes, and biosynthetic gene clusters (BGCs) in about 45 wildlife species sampled in a zoo environment. The data derived from captive wildlife species were compared to published data from free-ranging wildlife species.

We appreciate this very accurate summary of our study. We feel that this short paragraph summarizes our manuscript better than we managed to state the objective ourselves. We took this summary thus as one part of our introduction section, where we clearly mention our hypothesis and ambition (please see also our reply to the last point of reviewer 1 and the third point of this reviewer).

The study accumulated interesting data that might provide the baseline for many important research questions. It assembled a large amount of species-level genome bins (SGBs), predicted complete BGCs, reported that the diet influence variations in metabolic pathways, as well as differences in the microbiome between captive and free-ranging wildlife. The species match was taxonomically as close as possible but not a pairwise comparison per species was feasible. Moreover, and I think most interestingly, the work unveiled AMR genes against common veterinary antibiotics and resistance to vancomycin, a critical antibiotic in human medicine.

Thanks for the positive assessment. AMR is indeed one of the interesting aspects. We thus improved the analyses in this regard in the revised manuscript even more. At the same time, we mention technical challenges in identifying AMRs especially from the wildlife data in the revision.

Though the study has many merits, and a large amount of lab and bioinformatic work has been invested, the results section lacks a clear flow of in-depth analyses in relation to clearly defined research questions. Clearly developed hypotheses and predictions in relation to already available knowledge are missing. It is not surprising (and new) that the microbiomes differ according to diet (herbivores-omnivores-carnivores), phylogeny, stool – saliva, and also between captive and free-ranging wildlife. But I am sure more elaborated questions could be asked.

While we started the project with a clear research question: “how is the microbial composition of many different animal species in a zoo, where animals are under a very controlled environment –

especially under controlled nutrition”, we got carried away while doing the bioinformatics analysis. This might actually not have been a problem if we would not have missed to state it clearly and to mention the main purpose of the study. Through the whole manuscript we now improve this, also improving the overall readability of the manuscript. Please see also our general cover letter, the very last point of reviewer 1 and the first point of this reviewer in this regard.

Very interesting are the findings about AMR in zoo animals to a common antibiotic in humans. Specifically, they observed resistance genes against vancomycin, which is a last resort antibiotic against infections with Gram-positive bacteria in human medicine.

We discuss this aspect in a more detailed manner in the discussion section of the revised manuscript due to the modification of this section.

I think the data are not sufficiently explored and might hide a wealth of information that could be investigated in more detail. I appreciate the data outline and indication of data quality in Fig. 1. Looking at the study design, with the (partly) unbalanced sample, methodological challenges and data output I suggest to develop more specific research questions. An important question could be to follow up the AMR observations. E.g., understanding the spread of AMR between species and humans according to contact probability and horizontal gene transfer. It is not only a zoo-relevant question to investigate advantages/disadvantages of animal co-housing and contact to humans on microbiota communities and health, in generating an increased or decreased microbial diversity, and especially the impact of horizontal gene transfer (AMR). Along this line, the Intro is quite broad and unfocused and should be streamlined.

From some perspectives the data are not sufficiently explored, while other parts might be even “over-explored”, but we obviously failed to demonstrate this in the original manuscript. As mentioned, we completely reworked the introduction and focused on the original research question. We remove much of the human introduction background and add more for the animals.

We agree that the spread between Humans and Animals is an interesting factor and we investigate this in greater detail. It is, however, fair to say that the evidence remains indirect (e.g. we checked for the presence of AMRs on plasmids). However, we provide future recommendations on future research needs in the revised Discussion. We have currently no IRB in place that allows for analyzing the human AMRs in the zoo setting. We remain confident to get this approval for a new study that includes several zoos.

Reviewer #3 (Remarks to the Author):

The manuscript deals with the microbiomes of several wild and captive animal species and compares also the differences between wild and captive for certain species. It is most interesting from a descriptive point of view, and less from an analytical point of view comparing the different animal species as for most species only one sample was included.

We thank you for the fair summary of the study. We agree that our ambition was to have a description of the homogeneity or heterogeneity of animals in a zoo. We acknowledge this factor even better in the revised manuscript.

While the introduction is focused on the differences in microbiomes by diet,... (line 41), however it could include also the functional profiles which in general differ less.

This is a very fair point. We completely reworked the introduction, not only in this but also in other related points. Especially, we focus more on the one health aspect in the revised manuscript.

It is a pity that for most animal species included only one sample is available and as such the intraspecies variation cannot be taken into account.

We cannot argue against this point. It is a weakness of the study that for most animals only one replicate is available. We mention this fact even more clearly in the revision and also state that our ambition is to extend the study significantly. This does not only include sampling at two or more zoos but also considering the human microbiota being in contact with the animals. This new study goes however beyond the scope of a revision of this study and will likely take 2 years to be completed, including proper IRB approval.

The inclusion of culture-based determination of the microbiota is certainly an addition to the knowledge of the microbial diversity. The comparison between saliva and stool samples is however not relevant as these are quite different microbiota. The description of that part is however quite limited and mainly included in the supplemental material. The use of mostly non-selective plates has probably also led to the limited number of different species isolated. It would have been good to include a larger diversity of plates for the isolation of species seen this is compared to the non-culture based methods. The comparison between herbivores, carnivores and omnivores is also not very relevant for the culture-based methods. Moreover, there may be differences between the age within an animal species. This is for example very well known for poultry. The statement on line 157 that only 30-60% is cultivable needs referencing. According to my information it is much less.

We thank the reviewer for their valuable comment. We agree that culture-based approaches are always limited by the selected culture media. We chose standard agar plate media to allow growth of the highest number of different species. If we used mainly selective media we might increase growth of certain bacteria, however, it would be difficult to set the limit of plates. As metagenomic sequencing is much more precise, we use the culture-based approach mainly as a quality control of the sample, for which some bacteria that are expected to be present do not grow if the sample underwent poor transportation or storage for example. Further, we compare grown bacterial species with what we see after sequencing. If we detected certain species by culturing, however not during sequencing we can identify the lacking information in especially databases used for taxonomic profiling. Therefore, we would not expand the culture-based analysis as it is mainly performed for quality control reasons.

Since we compared the different diets briefly for the sequencing results, we also looked for similarities or differences within the culture-based results. However, we agree there are certainly more aspects we could look at. As we are mainly limited to one or two animals per animal species it is difficult to compare age for example. This would be possible only for a limited number of samples and would create groups of two or three individuals. We agree that this is indeed an interesting fact to explore, however we would need to increase the amount of individuals per animal species to draw conclusions here. We also changed the statement about 30-60 % cultivatable bacterial species and used the wording "It is worth mentioning that not every microorganism is cultivatable under laboratory conditions, making the metagenomic analysis a more powerful and more precise tool to investigate the microbiome." We decided to go with a rather brief description, as it is difficult to get a grasp of the real numbers since that would mean to try to culture every bacterium that exists. The "1% culturability paradigm" is not necessarily proven yet, for which we hope the reviewer is satisfied with the respective change in the manuscript.

The part on antimicrobial resistance is not clear. It is not clear what is detected in general and how the analysis was made, was the focus on acquired resistance genes only or has a more general database as CARD being used for the assessment? Different databases may give quite different results. Showing the

data as resistant against a certain antimicrobial is not that helpful in understanding the data. It is unclear why the specific focus is on vancomycin resistance. VanO has moreover been described in elephants and thus has been detected before in animal samples. The part on the transfer of resistance within the manuscript as this needs a very different methodology to assess. It should be noted that in all ecosystems antimicrobial resistance genes are found.

Thank you for your comment. We substantially improved the analysis of AMR genes in our dataset by using three different tools (AMRFinderPlus, DeepARG, Resfinder). Our exact methods are documented in line 467 ff. By doing so, we now provide a more comprehensive appraisal of the antimicrobial resistance spectrum. Indeed, we found considerable rates of resistance against commonly used antibiotic classes such as tetracyclines, macrolides, lincosamides, but also certain beta-lactamases. Surprisingly, we found resistances against vancomycin in primarily little captive monkeys, which are usually in close contact to zoo keepers, making a transfer of vancomycin resistant bacteria from animal to human possible. We agree that antimicrobial resistance genes can be found in all ecosystems, highlighting the need to monitor not just bacteria colonizing humans, but also the environment and animals, to ensure an overall brought AMR surveillance and enable the detection of possible transmission routes.

References

1. Lim, S.J. and Bordenstein, S.R. (2020) An introduction to phyllosymbiosis. *Proc Biol Sci*, **287**, 20192900.
2. Feldgarden, M., Brover, V., Gonzalez-Escalona, N., Frye, J.G., Haendiges, J., Haft, D.H., Hoffmann, M., Pettengill, J.B., Prasad, A.B., Tillman, G.E. *et al.* (2021) AMRFinderPlus and the Reference Gene Catalog facilitate examination of the genomic links among antimicrobial resistance, stress response, and virulence. *Sci Rep*, **11**, 12728.
3. Arango-Argoty, G., Garner, E., Pruden, A., Heath, L.S., Vikesland, P. and Zhang, L. (2018) DeepARG: a deep learning approach for predicting antibiotic resistance genes from metagenomic data. *Microbiome*, **6**, 23.
4. Florensa, A.F., Kaas, R.S., Clausen, P., Aytan-Aktug, D. and Aarestrup, F.M. (2022) ResFinder - an open online resource for identification of antimicrobial resistance genes in next-generation sequencing data and prediction of phenotypes from genotypes. *Microb Genom*, **8**.
5. Rehner, J., Schmartz, G.P., Groeger, L., Dastbaz, J., Ludwig, N., Hannig, M., Rupf, S., Seitz, B., Flockerzi, E., Berger, T. *et al.* (2022) Systematic Cross-biospecimen Evaluation of DNA Extraction Kits for Long- and Short-read Multi-metagenomic Sequencing Studies. *Genomics Proteomics Bioinformatics*, **20**, 405-417.

Reviewers' Comments:

Reviewer #2:

Remarks to the Author:

I appreciate the large effort of the authors in revising their work. Many missing details especially in the Method section are now incorporated, and clarified. The study acknowledges now all shortcomings (single sample per species, not perfect species match between captive and wildlife samples, use of fecal samples – saliva, etc.), and provides clear Suppl Tables and Figures for assessment. I also agree with the new title “Exploring Microbial Diversity and Biosynthetic Potential in Zoo and Wildlife Animal Microbiomes” since it fits much better the content of the MS.

Nevertheless, despite all these improvements, the study remains largely descriptive and as the title correctly summarizes – explorative. I appreciate that the work provides an important summary of information, but the general conclusions that animal microbiomes vary according to diet, and husbandry etc. are not new. Interesting is the AMR part though.

Specific comments

Intro: still lacks a V-shaped structure. Each paragraph should be streamlined regarding its message/contribution to the overall MS focus. For example, the first paragraph finishes with the sentence “However, wildlife animals can travel major distances and interact with other animals through inter-species or intra species interactions, providing numerous opportunities to acquire and spread AMR”. Ok, but this has nothing to do with your study set-up comparing zoo and wildlife species. You detect higher AMR in zoo animals (that don’t travel at all)...

The second and third paragraphs are also very general, and the reader does not get why the comparison between zoo and wildlife provides new information that has not been gained before.

The MS would benefit from clearly formulated research questions that are then clearly addressed in the results – moving beyond characterization / exploring, towards a hypothesis-driven approach with a justified prediction in mind.

Reviewer #3:

Remarks to the Author:

The remarks have been replied appropriately. I do not have any further comments.

RESPONSE TO REVIEWERS' COMMENTS

Reviewer #2 (Remarks to the Author):

I appreciate the large effort of the authors in revising their work. Many missing details especially in the Method section are now incorporated, and clarified. The study acknowledges now all shortcomings (single sample per species, not perfect species match between captive and wildlife samples, use of fecal samples – saliva, etc.), and provides clear Suppl Tables and Figures for assessment. I also agree with the new title “Exploring Microbial Diversity and Biosynthetic Potential in Zoo and Wildlife Animal Microbiomes” since it fits much better the content of the MS.

We would like to thank the reviewer once again for this and previous rounds of feedback, which have helped us enhance the quality of our work.

Nevertheless, despite all these improvements, the study remains largely descriptive and as the title correctly summarizes – explorative. I appreciate that the work provides an important summary of information, but the general conclusions that animal microbiomes vary according to diet, and husbandry etc. are not new. Interesting is the AMR part though.

We agree with the reviewer that the mentioned individual points may not be considered novel and have already been highlighted in better-controlled environments, such as dedicated animal models. Nevertheless, we believe that confirmation of existing knowledge should blend with new insights as a step of quality control for our own data, as well as independent validation of existing research. Additionally, we believe that the newly gained insights into BGCs and AMR will be of great service to the research community. While, as rightfully stated, these results are more descriptive in nature, we still believe in the value and future potential of these findings.

Specific comments

Intro: still lacks a V-shaped structure. Each paragraph should be streamlined regarding its message/contribution to the overall MS focus. For example, the first paragraph finishes with the sentence “However, wildlife animals can travel major distances and interact with other animals through inter-species or intra species interactions, providing numerous opportunities to acquire and spread AMR”. Ok, but this has nothing to do with your study set-up comparing zoo and wildlife species. You detect higher AMR in zoo animals (that don't travel at all)...

The second and third paragraphs are also very general, and the reader does not get why the comparison between zoo and wildlife provides new information that has not been gained before.

We may agree with the reviewer that our introduction reads a little broad and especially the first two paragraphs can appear detached from the manuscript's main focus. Nevertheless, we believe that we do have a rapid convergence to our manuscript's focus in the third paragraph allowing us to introduce the problem broadly initially and provide more context. The first paragraph has the goal of highlighting the importance of animal microbiomes in the

context of AMR spread. The second paragraph is mostly focused around BGCs in animal microbiomes that potentially are capable of synthesizing novel antimicrobial compounds. Only the third paragraph then highlights the main stand-alone properties of our dataset which are intertwined with the zoo as a sampling environment. The same paragraph also highlights zoo animals as the intermediary animal model between wild and farm/ laboratory animals.

The MS would benefit from clearly formulated research questions that are then clearly addressed in the results – moving beyond characterization / exploring, towards a hypothesis-driven approach with a justified prediction in mind.

We agree with the reviewer that a clearly formulated hypothesis would be beneficial for the flow of the manuscript and is of course central to good scientific practice. However, we firmly stand behind the explorative nature of this specific piece of work. We are convinced that in combination with another piece of work we are soon going to publish, we, as well as other parts of the community, will be able to synthesize new interesting hypotheses focused around host-derived BGCs.

Reviewer #3 (Remarks to the Author):

The remarks have been replied appropriately. I do not have any further comments.

We thank the reviewer for all the feedback they provided.